

# The test-retest reliability of a Body Posture Literacy Questionnaire among Polish teachers from different educational levels

Marta Kinga Labecka[1], Agnieszka Jankowicz-Szymańska[2], Magdalena Plandowska[3], Elżbieta Olszewska[4] and Reza Rajabi[5]

[1] Faculty of Rehabilitation, Jozef Pilsudski University of Physical Education, Warsaw, Poland
[2] Department of Physiotherapy, Faculty of Health Sciences, University of Applied Sciences in Tarnow, Tarnow, Poland
[3] Faculty of Physical Education and Health in Biala Podlaska, Jozef Pilsudski University of Physical Education in Warsaw, Biala Podlaska, Poland
[4] Jozef Pilsudski University of Physical Education, Warsaw, Poland
[5] Department of Health and Sport Medicine, Faculty of Sport Sciences and Health, University of Tehran, Tehran, Iran

Corresponding author
Marta Kinga Labecka,
marta.labecka@awf.edu.pl

## ABSTRACT

**Background**. The study aimed to develop a reliable and valid Teachers' Body Posture Literacy Questionnaire (TBPLQ) to examine their body posture knowledge.

**Methods**. The tool was based on a Parents Body Posture Literacy Questionnaire (PBPLQ) and modified and validated through discussion with experts, conducted in two rounds. Corrective gymnastics, physiotherapy, ergonomics, and physical education (PE) experts and doctoral and postdoctoral scholars evaluated content validity. Test-retest repeatability was tested using Cohen's kappa coefficient. The study used a convenience sample of 195 teachers from three different educational levels: PE, kindergarten, and primary education in two rounds of test-retest (pilot test-retest reliability and main test-retest reliability of the questionnaire). The first round encompassed 95 participants, with pre-test and post-test procedures applied using the original TBPLQ. The second round involved 100 participants and followed a similar approach, incorporating modifications to the TBPLQ based on the reliability outcomes observed in the first round.

**Results**. The results of the first-round test-retest TBPLQ reliability, with 95 samples, resulted in an overall reliability of 0.77 (range 0.02 to 1). This indicated that the questionnaire still lacks sufficient reliability. Consequently, after the necessary amendments and modifications, the questionnaire's reliability was tested for the second time with 100 samples. Notably, the overall reliability of 0.82 (ranging from 0.50 to 1) was established for the TBPLQ indicating that 87.5% of the questionnaire items achieved reliability scores within the substantial and almost perfect range and only 12.5% of the items attained moderate reliability scores.

**Conclusions**. The questionnaire is a new self-report measure for evaluating teachers' literacy in postural health. It is applicable in both research and practical contexts, extending its use to larger and more diverse populations.

## INTRODUCTION

Postural defects, a contemporary pandemic and societal issue (*Kolarová et al., 2019*), affect 60–80% of children (*Klimkiewicz-Wszelaki et al., 2019*). Causes include congenital or acquired defects, exogenous factors (physical activity, eating habits, body position), and endogenous factors affecting general health (*Modrzejewska & Malec, 2017*). Noteworthy environmental factors involve prolonged sitting in classrooms, poor posture, inadequate furniture (*Murphy, Buckle & Stubbs, 2004*), and the manner of carrying school backpacks (*Minghelli, Nunes & Oliveira, 2021*). An inappropriate lifestyle and insufficient physical activity stand out as common causes (*Skorupka & Asienkiewicz, 2014*). Preventive measures are vital given the risk of body posture defects at every child development stage. Early diagnosis, especially in children under 15, is crucial for skeletal modification and easier postural correction (*Mitova, 2015*). Untreated poor posture can lead to static body impairments, musculoskeletal disorders, pain, increased healthcare costs, and mental symptoms (*Domljan, Vlaović & Grbac, 2010*; *Hagner, Bak & Hagner-Derengowska, 2010*). Untimely correction significantly impacts life quality (*Kedra et al., 2021*). The situation highlights the need to monitor children's locomotor apparatus development systematically during early school years and to take preventive action from early childhood (*Kowalski, Kotwicki & Siwik, 2013*). Preventive actions should include effective screening tests (*Lipkin et al., 2020*; *Moment et al., 2021*). Responsibility for supporting child development extends beyond parents to those shaping the child's environment (*Supreme Audit Office, 2020*).

In addition to specialists (family or general practitioner, pediatrician, nurse), in cooperation with parents, the school should be the first environment for preventing postural issues, which involves eliminating factors that negatively affect the development of the child's body (*Skorupka & Asienkiewicz, 2014*). The school is responsible for creating conditions for the child's functioning to prevent the formation and spread of body posture defects. Primary education teachers play a crucial role, as the first three years of schooling contribute to posture deterioration (*Nichele, Turra & Badaró, 2016*). The teachers can observe changes and ensure a correct working environment. Teachers should shape correct sitting habits and posture (*Gao et al., 2021*). Physical Education (PE), as a fundamental component of engaging in physical culture, significantly contributes to molding the character of young individuals. In this context, educators must acknowledge that PE facilitates holistic development, instills a sense of joy through physical effort, and supports overall health maintenance (*Ruzimbaevich & Ruzimbaev, 2021*). PE teachers also contribute significantly to preventing body posture defects (*Khakimovich & Rozmatovich, 2022*). Undoubtedly, teachers' knowledge about the role of correct body posture and how to create it, as well as appropriate motivation for this type of action, skills, and preventive action is of great importance in students' postural education.

Although some studies have analyzed knowledge of body posture and its prevention, there are limitations. Few studies examined parents' knowledge of body posture but

the assessment tool was not made available (*Jankowicz-Szymanska, Nowak & Slomski, 2010*; *Klimkiewicz-Wszelaki et al., 2019*; *Ryabova et al., 2021*; *Rajabi et al., 2023*). Few self-reported evaluation instruments were developed to study body postural habits in adolescent populations or college students (*Noll et al., 2013*; *Monfort-Pañego & Miñana Signes, 2020*; *Feng & Zhang, 2023*). *Salman et al. (2024)* analyzed various questionnaires to assess the knowledge regarding postural behavior and the prevention of defects of body posture. They revealed that 21 questionnaires had been developed to assess knowledge regarding postural habit, ergonomics, and posture. Most of these tools were designed for university students or schoolchildren, were not properly validated and their value is moderate. To the best of the researcher's knowledge, no scientific studies exist on educational teachers' knowledge regarding prevention, correction, or the causes of body posture defects. As a result, this study aims to develop a valid and reliable questionnaire examining the literacy of teachers in postural health, including postural defeat recognition in pupils and its probable causes and solutions.

## MATERIALS & METHODS

### Ethics

The study was conducted by the Declaration of Helsinki and approved by the Ethics Committee of Jozef Pilsudski University of Physical Education in Warsaw (protocol code 01-07/2023, date of approval 18 February 2023). Participation in the study was voluntary. Respondents provided written informed consent, and data privacy and confidentiality were assured by treating the data aggregately and coding the names.

### Stages of developing the Teachers' Body Posture Literacy Questionnaire

This research included four phases. Phase 1 was from February to March 2023 and involved creating a questionnaire based on the Parents Body Posture Literacy Questionnaire (PBPLQ) (*Labecka et al., 2024*) and adjusted by expert consultation. Phase 2 was conducted in April 2023. The questionnaire was translated into Polish. Phase 3 was conducted in August 2023, as a first round test-retest reliability (pilot test-retest reliability). Phase 4 was a survey to test the questionnaire's reliability of the revised questionnaire in November 2023 as second round test-retest reliability (main test-retest reliability). This was done after revisions were made to the English version of the questionnaire based on the results of the first round test-retest reliability study (Fig. 1), by the international committee of experts.

### Questionnaire development

#### Validity assessments (content and construct validity)

The questionnaire was based on a pool of items from previous questionnaires (*Labecka et al., 2024*; *Rajabi et al., 2023*). Most questions were obtained from the PBPLQ (*Labecka et al., 2024*). This questionnaire was chosen because it was highly reliable and relevant in assessing postural knowledge. In addition, it was scientific and was prepared according to rigorous questionnaire development procedures.

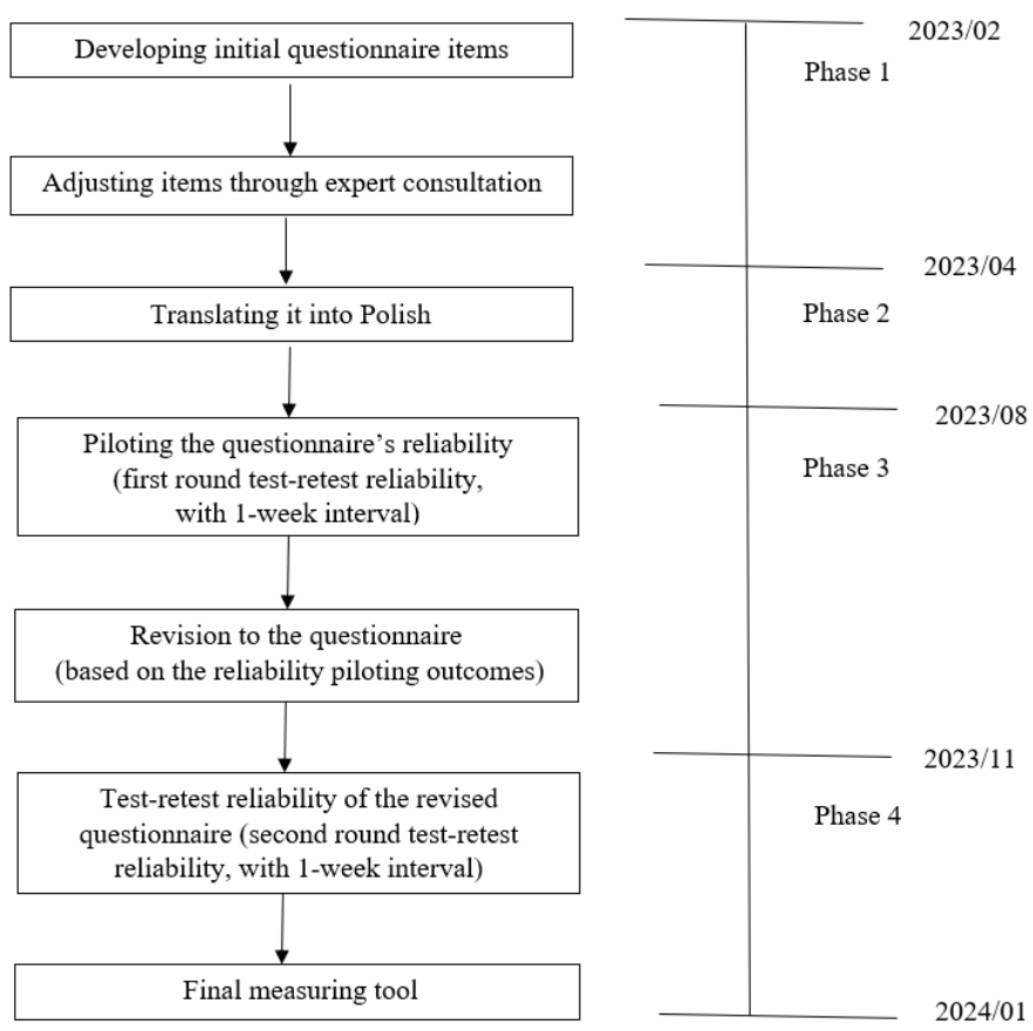

**Figure 1 Phases of developing the Teachers' Body Posture Literacy Questionnaire (TBPLQ).**

A committee of six experts in postural health (doctoral and postdoctoral scholars, physiotherapists, PE teachers, and instructors of corrective gymnastics/exercises) evaluated the draft questionnaire in English for content validity. Based on literature from postural health and discussions between experts, questions were removed or reworded and new ones were added. Because the draft questionnaire was aimed at parents, the questions were modified for other audiences, including teachers. The initial question pool contained 36 items divided into three sections: postural abnormality, postural ergonomics, and sociodemographic data. Working independently, the experts considered which questions were most important and assessed the questionnaire. Then they discussed the formulation of the questions and answers and the structure and format of the questionnaire in face-to-face and online meetings. A brainstorming session was conducted. The committee of experts was international, so discussions were conducted in English. During multiple exchanges of information and feedback revision, the Delphi method's elements (*McPherson,*

*Reese & Wendler, 2018*) transformed the experts' opinions into a group consensus. Final recommendations were taken from the arguments that were most widely accepted by the participants to improve the questionnaire. The original questionnaire was developed in English. The pool of questions turned out to be much larger than the final number of items that were included in the PBPLQ model questionnaire.

### Translation of the questionnaire

After determining the preliminary version of the questionnaire, it was translated from English into Polish using the guidelines recommended by Oxford University Innovation to ensure appropriate standards (*Wild et al., 2005*). Forward translation by two independent translators (native Polish speakers), a discussion between 'forward' translators and the Polish research team, backward translation by two independent translators (native English speakers), and a review by a physiotherapist established the final Polish version of the questionnaire. The authors have permission from the copyright holders to use this instrument.

## Reliability analysis
### Pilot test-retest reliability

The first round of test-retest reliability as pilot reliability was conducted twice, with approximately 1 week between the first and second administration of the questionnaire to obtain information on how the Teachers' Body Posture Literacy Questionnaire (TBPLQ) questionnaire was working, whether it was feasible to use it in a real-world setting, and to complete it in a reasonable amount of time, assessing its ease of understanding, its acceptability and evaluating its initial test-retest reliability. At that time, the respondents were not allowed to see the answers they gave when the questionnaire was first administered. The authors chose the interval between questionnaire administrations based on the concept that the time should be too long to allow respondents to remember previously selected answers, but not long enough to allow respondents to acquire new knowledge (*Parmenter & Wardle, 2000*). The questionnaire was administered to 95 teachers of both sexes and of different ages. Respondents were recruited from all school types (kindergarten, primary, secondary, and high school) in various Polish cities. The selection process for Polish teachers involved a stratified sampling approach to include participants from primary, kindergarten, and PE levels. This method ensured that the sample represented the broader educational spectrum. Data were collected *via* an online questionnaire distributed through educational forums and social media (emails, Facebook posts, *etc.*) including a link to an online preliminary version of the questionnaire. A backend researcher checked the electronic surveys, and invalid surveys were eliminated. Respondents were invited to give their opinions of the questionnaire, their difficulties in understanding the questions and answers, and comment on other aspects. Owing to slight test-retest reliability results in some of the questions, and the opinions of the experts and participants, the questionnaire was modified.

*Main test-retest reliability*

Finally, the experts decided to carry out the test-retest again to check the reliability of the modified questionnaire. A second round of test-retest reliability was performed with 100 newly enrolled teachers who were unfamiliar with the content and structure of the questionnaire. Participants were recruited to validate the modified questionnaire in the same way as a first round test-retest, performing test-retest reliability with a 1-week interval. This sequential approach ensured the reliability of the initial instrument and the validity of the revised version.

## Statistical analysis

Data were analyzed using Microsoft Excel and were presented as the range, mean, or *n* (%). Cohen's kappa coefficient was applied to evaluate the test-retest reliability (data from the same rater at two different points in time). Values $\leq 0$ indicating no agreement, 0.01–0.20 as none to slight, 0.21–0.40 as fair, 0.41–0.60 as moderate, 0.61–0.80 as substantial, and 0.81–1.00 as almost perfect agreement (*McHugh, 2012*). To ensure the content validity of the questionnaire, experts from various disciplines and different international institutions reviewed and agreed upon the questionnaire's content coverage. Rigorous data collection protocols were implemented, including standardized instructions and pilot testing of the survey instrument, to ensure transparency and consistency. This approach allowed for gathering reliable data that accurately reflected the teachers' perspectives on body posture knowledge/literacy across different teaching stages.

## RESULTS

### Questionnaire development
*Validity assessments (content and construct validity)*

The initial questionnaire was based on previous questionnaires for parents and included a total of 41 items. Based on expert opinion, in the first part of the questionnaire (postural abnormality), a fourth column was added containing graphics presenting corrective exercises for the abnormality. Also, in the third column (causes of the abnormality) an answer combining two or three correct answers was added (*e.g.*, 1&2 or 1,2,3). In the second part of the questionnaire (postural ergonomics), an 'All of the above' option was added to the second column. The third part of the questionnaire was significantly modified (the question pool was increased from 20 to 25 items) to include teachers' sociodemographics, professional data, and educational status.

### Reliability analysis
*Characteristics of questionnaire respondents*

Table 1 presents the sociodemographic and professional characteristics of the teachers. Ninety-five teachers completed the first questionnaire administered (I round), where the majority were female (76.8%). Respondents aged 26–58 (38.20 ± 10.97), came from different educational levels and had over 10 years of work experience (11.59 ± 10.49). It was necessary to modify some questions and sub-questions to improve them based on respondents' comments.

**Table 1  Sociodemographic and professional characteristics of questionnaire respondents.**

| Item | | I round | | II round | |
|---|---|---|---|---|---|
| | | **n** | **%** | **n** | **%** |
| Sex | Female | 73 | 76.8 | 58 | 58.0 |
| | Male | 22 | 23.2 | 42 | 42.0 |
| Age (years) | ≤ 35 | 20 | 21.1 | 59 | 59.0 |
| | 36–45 | 47 | 49.5 | 32 | 32.0 |
| | ≥ 46 | 28 | 29.4 | 9 | 9.0 |
| Education level | Kindergarten | 33 | 34.7 | 32 | 32.0 |
| | Primary school | 30 | 31.6 | 29 | 29.0 |
| | Physical education | 32 | 33.7 | 39 | 39.0 |
| Work experience | ≤ 10 | 19 | 20.0 | 42 | 42.0 |
| | 11–20 | 55 | 57.9 | 39 | 39.0 |
| | ≥ 21 | 21 | 22.1 | 19 | 19.0 |

**Notes.**

n, number of participants;  %, percent.

The 100 teachers, of whom the majority were female (58.0%), filled out the modified version of the questionnaire (II round). Respondents aged 25–48 years (30.24 ± 8.57) came from different educational levels with under 10 years of work experience (8.54 ± 6.46). There was no adjustment to the questionnaire after the second round reliability study as the modifications (content and graphics) after the second round were effective and in line with expert group expectations.

### Pilot test-retest reliability

Table 2 presented the Cohen's kappa coefficient results for all items in the teachers' groups included in the study. After the first round of test-retest reliability (95 teachers out of 108), the reliability scores ranged from 0.02 to 1.00 overall, 0.74 for the postural abnormality questions, and 0.80 for the postural ergonomics questions. In the first part of the questionnaire (postural abnormality items), sub-item 2 from question 1 (Cohen's kappa = 0.02), sub-item 2 from question 2 (Cohen's kappa = 0.37), sub-item 3 from question 5 (Cohen's kappa = 0.54), and sub-item 3 from question 8 (Cohen's kappa = 0.46) obtained slight reliability scores. In the two sections, sub-item 1 from question 9 (Cohen's kappa = 0.53), sub-item 2 from question 11 (Cohen's kappa = 0.55), and sub-item 1 from question 13 (Cohen's kappa = 0.59) displayed moderate reliability. After analyzing the respondents' answers, the researchers observed a lack of clarity resulting in multiple interpretations of selected questions, so modifications to the questionnaire were required. In the first part of the questionnaire, in column 3, the causes of abnormalities were reformulated into statements describing the abnormalities; in column 4, the answer "I don't know" was added, in column 2, in question 1, 2, the names of the abnormalities were changed; in column 3, in question 5,8, statements describing the abnormalities were modified. The second part of the questionnaire (postural ergonomics items) remained unchanged. In the third part of the questionnaire (sociodemographics and professional characteristics), one question about external and internal factors affecting body posture

was deleted, and question 13 was modified. Finally, the total number of questions was changed (from 41 to 40).

### Main test-retest reliability

In the second round of test-retest reliability of TBPLQ, (100 teachers out of 118) were assessed using 16 questions with 48 sub-questions, 32 sub-questions related to postural abnormalities, and 16 sub-questions related to postural ergonomics. The overall reliability (16 questions) of the TBPLQ was demonstrated as 0.82 (range 0.46 to 1.00), with 54.1% demonstrating almost perfect reliability (range 0.81 to 1.00), 31.2% demonstrating substantial reliability (range 0.61 to 0.80) and 14.6% demonstrating moderate reliability (range 0.41 to 0.60). More detailed results regarding reliability for postural abnormalities and postural ergonomics are presented in Table 2.

## Final questionnaire

After refinements and modifications were made because of the validation process and the first round of reliability tests, the final questionnaire (the validated Polish version, and the English version) was formulated (Supplementary material). The second round of reliability of TBPLQ indicated a substantial improvement. The first and second round reliability test comparison is provided in Fig. 2. Finally, the questionnaire took the form of a self-administered document that was easy to complete and not time-consuming; it consisted of three parts (40 questions –including eight questions for postural abnormality with 32 sub-items, eight questions for postural ergonomics with 16 sub-items, and 24 questions about sociodemographics, professional data, and educational status).

## DISCUSSION

The primary aim of this study was to assess the test-retest reliability of a self-reporting questionnaire designed to evaluate the level of teachers' literacy in postural health, encompassing aspects such as recognition, naming, describing, selecting an appropriate corrective exercise for postural abnormalities, and the ergonomics of daily activities.

The results of the first round test retest TBPLQ reliability resulted in overall substantial reliability. However, some questionnaire items showed slight to moderate reliability, which indicated that the questionnaire still lacked sufficient reliability. Upon closer examination of sub-items 9 and 13, it became apparent that they contained misleading and ambiguous elements. For example, in the first sub-item of item 9, variations in graphic patterns (some depicting a child and others an adult) were identified. Similarly, the first sub-items of item 13 presented graphics with striking similarities, making it challenging to select the correct response. It is important to note that certain refinements were necessary to enhance reliability and to address the observed discrepancies between expected and actual outcomes, so specific modifications to the final questionnaire were made. These adjustments included redesigning specific items and their sub-items based on experts' and participants' opinions. Consequently, after the necessary amendments and modifications, the questionnaire's reliability was tested for the second time. Our study showed that the overall test-retest reliability rates for the TBPLQ were substantial, with 82% of the items achieving reliability

Labecka et al. (2024), *PeerJ*, DOI 10.7717/peerj.17952

**Peer**J

**Table 2  Data describing the reliability of questionnaire items.**

| Question/item (Postural abnormality) | | Sub-item1 (Recognition of abnormalities) | | Sub-item2 (Naming of the abnormality) | | Sub-item3 (Causes of the abnormality) | | Sub-item4 (Corrective exercise for the abnormality) | | Items average (range, mean) | |
|---|---|---|---|---|---|---|---|---|---|---|---|
| | | 1 round | 2 round | 1 round | 2 round | 1 round | 2 round | 1 round | 2 round | 1 round | 2 round |
| 1 | Forwarded head | 1 | 1 | 0.02 | 0.82 | 0.70 | 0.67 | 0.83 | 0.68 | 0.02–0.83 (0.64) | 0.67–1 (0.79) |
| 2 | Thoracic Kyphosis | 1 | 1 | 0.37 | 0.69 | 0.80 | 0.73 | 0.81 | 0.56 | 0.37–0.81 (0.75) | 0.56–1 (0.75) |
| 3 | Lumbar Lordosis | 0.85 | 1 | 0.83 | 0.81 | 0.83 | 0.52 | 0.83 | 0.67 | 0.83–0.85 (0.84) | 0.52–1 (0.75) |
| 4 | Scoliosis | 0.90 | 1 | 0.78 | 1 | 0.62 | 1 | 0.74 | 0.89 | 0.62–0.90 (0.76) | 0.89–1 (0.97) |
| 5 | Geno Varum | 0.75 | 1 | 0.64 | 0.74 | 0.54 | 0.73 | 0.73 | 0.76 | 0.54–0.75 (0.67) | 0.73–1 (0.81) |
| 6 | Geno Valgum | 0.91 | 1 | 0.85 | 0.5 | 0.82 | 0.79 | 0.61 | 0.7 | 0.61–0.91 (0.80) | 0.50–1 (0.75) |
| 7 | Feet abnormalities | 0.73 | 0.81 | 0.66 | 0.81 | 0.64 | 0.55 | 0.68 | 0.76 | 0.64–0.73 (0.67) | 0.55–0.81 (0.73) |
| 8 | Toe abnormalities | 1 | 1 | 0.85 | 0.95 | 0.46 | 1 | 0.71 | 0.75 | 0.46–1 (0.76) | 0.46–1 (0.92) |
| **Sub-items average** | | **0.73–1.00 (0.89)** | **0.81–1 (0.98)** | **0.02–0.85 (0.63)** | **0.50–1 (0.79)** | **0.46–0.83 (0.68)** | **0.52–1 (0.75)** | **0.61–0.83 (0.74)** | **0.56–0.89 (0.72)** | **Total = 0.74** | **Total = 0.81** |

| Postural Ergonomics Questions/items | | Sub-item1 Cohen's kappa value (Recognition) | | Sub-item2 Cohen's kappa value (Mechanism) | | | |
|---|---|---|---|---|---|---|---|
| | | 1 round | 2 round | 1 round | 2 round | 1 round | 2 round |
| 9 | Sitting posture on the floor while playing | 0.53 | 0.7 | 0.77 | 0.84 | 0.53–0.77 (0.65) | 0.70–0.84 (0.77) |
| 10 | Lifting and carrying postures | 0.83 | 1 | 0.78 | 0.5 | 0.78–0.83 (0.81) | 0.50–1 (0.75) |
| 11 | Working with correct posture while working with computer | 1 | 1 | 0.55 | 0.6 | 0.55–1 (0.78) | 0.60–1 (0.8) |
| 12 | Walking posture | 1 | 1 | 0.78 | 0.7 | 0.78–1 (0.89) | 0.70–1 (0.85) |
| 13 | Smart phone using posture | 0.59 | 1 | 0.70 | 0.73 | 0.59–0.70 (0.65) | 0.73–1 (0.86) |
| 14 | Writing posture | 1 | 1 | 0.85 | 0.58 | 0.85–1 (0.93) | 0.58–1 (0.79) |
| 15 | Back pack carrying posture | 1 | 1 | 0.78 | 0.97 | 0.78–1 (0.90) | 0.97–1 (0. 98) |
| 16 | Sleeping posture | 0.76 | 0.81 | 0.80 | 0.89 | 0.78–0.80 (0.78) | 0.81–0.89 (0.85) |
| **Sub-items average** | | **0.53–1.00 (0.84)** | | **0.55–0.85 (0.75)** | | **Total = 0.80** | **Total = 0.83** |
| | | **Over all 16 items** | | | | **0.77** | **0.82** |

**Notes.**
Data presented as range and mean; n –number of participant.

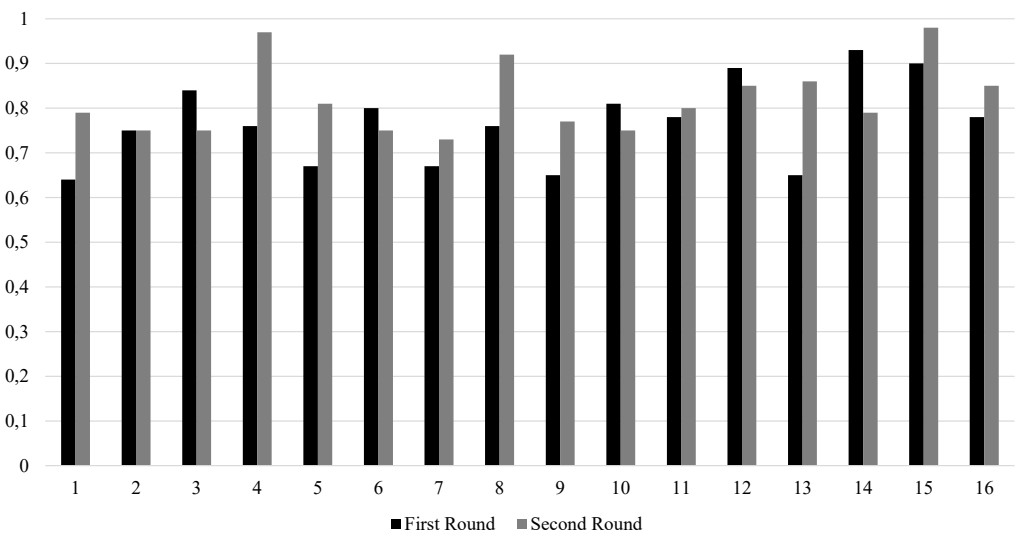

**Figure 2**   **Item by item reliability improvement of the TBPLQ.**

scores between 0.46 and 1.00. However, 14.6% of the items achieved moderate reliability scores ranging from 0.41 to 0.60. This nuanced assessment aligns with the findings of a recently published study (*Rajabi et al., 2023*).

Posture defects should be detected as early as possible. It was therefore indicated that school should be the first environment for preventing postural issues (*Skorupka & Asienkiewicz, 2014*). However, there are no questionnaires to measure body postural habits in young children and research teacher's knowledge regarding students' postural abnormalities. Therefore, we created such a tool for teachers who, at various stages of education, are particularly responsible for caring for the quality of the body posture and healthy lifestyle of children and youth. The analysis of the questionnaire results can be used to improve higher education programs to better prepare teachers to screen children for the occurrence of faulty body posture. For the teachers and educators working with this tool, it can be a pretext for self-assessment of knowledge about posture deformities. These teachers will also be inspired to provide children and adolescents with the knowledge necessary to develop the habit of taking care of their body posture throughout their lives. As a consequence, children and young people will receive better prevention of postural defects and deformities.

One limitation of this study is the sample size. The number of study participants was the minimum necessary to achieve reliable outcomes. In contrast, the study's strengths are rigorous questionnaire development involving multiple rounds of pretests, meticulous item selection, extensive expert reviews to ensure the content validity and clarity of the questionnaire and comprehensive content discussion of all the issues of the questionnaire.

## CONCLUSIONS

The Teachers' Body Posture Literacy Questionnaire (TBPLQ) demonstrated commendable overall reliability, encompassing substantial and almost perfect scores, signifying its potential to assess teachers' knowledge of postural health in school-age students effectively. The TBPLQ is a newly established self-report questionnaire, proven to be both valid and reliable, making it suitable for application in both research and practical settings. Its adaptability extends its utility to broader and more diverse populations.

## ACKNOWLEDGEMENTS

The authors would like to show gratitude to Mr. Piotr Tabor from Jozef Pilsudski University of Physical Education in Warsaw who helped collect the survey for the second round of pretest.

### Funding

This work was supported by the Ministry of Education and Science in the years 2023-2024 under the University Research Project at Józef Piłsudski University of Physical Education in Warsaw 'The knowledge of teachers about the prevention, correction, and defects of body posture in children' [280/R/PRO/2023; date 2023/02/28]. The funders had no role in study design, data collection and analysis, decision to publish, or preparation of the manuscript.

### Grant Disclosures

The following grant information was disclosed by the authors:
The Ministry of Education and Science in the years 2023-2024 under the University Research Project at Józef Piłsudski University of Physical Education in Warsaw 'The knowledge of teachers about the prevention, correction, and defects of body posture in children': 280/R/PRO/2023; date 2023/02/28.

### Competing Interests

The authors declare there are no competing interests.

### Author Contributions

- Marta Kinga Labecka conceived and designed the experiments, performed the experiments, analyzed the data, prepared figures and/or tables, authored or reviewed drafts of the article, and approved the final draft.
- Agnieszka Jankowicz-Szymańska conceived and designed the experiments, performed the experiments, analyzed the data, authored or reviewed drafts of the article, and approved the final draft.
- Magdalena Plandowska performed the experiments, authored or reviewed drafts of the article, and approved the final draft.
- Elżbieta Olszewska conceived and designed the experiments, performed the experiments, authored or reviewed drafts of the article, and approved the final draft.

- Reza Rajabi conceived and designed the experiments, performed the experiments, analyzed the data, authored or reviewed drafts of the article, and approved the final draft.

## Human Ethics

The following information was supplied relating to ethical approvals (i.e., approving body and any reference numbers):

The study was conducted by the Declaration of Helsinki and approved by the Ethics Committee of Jozef Pilsudski University of Physical Education in Warsaw (protocol code 01-07/2023, date of approval 18 February 2023).

## Data Availability

The raw measurements are available in the Supplemental File.

## Supplemental Information

Supplemental information for this article can be found online at http://dx.doi.org/10.7717/peerj.17952#supplemental-information.

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
