# Peer review of "The test-retest reliability of a Body Posture Literacy Questionnaire among Polish teachers from different educational levels"

_PeerJ, doi:10.7717/peerj.17952_

## Round 0.1 · original submission · Major Revisions

The authors must be commended for having conducted a difficult task. For this reason, I am willing to offer the authors a chance to adress the major shortcomings of the article pointed out by the 2 reviewers, but I am underscoring the ***major*** nature of the revisions required.

The first issue is twofold: the article need extensive editing for English language and cleaning the article out (surface/AI-driven editing will not be sufficient) and at the same time, needs restructuring

- the introduction is listwise, unstructured, and does not cover the literature in enough clarity
- the methods and results sections are not satisfactory at present. I think, but the decision is yours of course, that this article calls for a non-traditional structure (not methods/results/discussion) and would benefit from sectioning following the methods required to design and validate a survey.

There are several methodological inconsistencies throughout. These need to be fully addressed at resubmission.

The practical applications need to be plainly flushed out for translation into practice.

Please respond to every point the reviewers made (not mine) in a structured manner.

Reviewer 1 ·

Basic reporting

ntroduction and Literature Review: The introduction section requires expansion to adequately frame the research within the existing literature on questionnaire use in educational settings, particularly those assessing body posture literacy. This review should highlight the relevance and necessity of developing and validating the Body Posture Literacy Questionnaire (BPLQ), citing previous studies and established frameworks.

Clarity and Structure: The manuscript should clearly separate the processes of questionnaire validation and data analysis. This separation will help in maintaining the focus and allowing readers to easily follow the progression of the research. Consider organizing the manuscript into distinct sections or chapters, each dedicated to one specific aspect of the study.

Experimental design

Methodological Framework: The methodology employed in the study must adhere to the standards recommended by the International Society for Pharmacoeconomics and Outcomes Research (ISPOR). This involves detailing the steps taken in the validation of the questionnaire, including pilot testing, reliability testing (like test-retest reliability), and validity assessments (such as content and construct validity).

Sample Selection and Data Collection: Describe the process of selecting Polish teachers from various educational levels and the rationale behind this choice. Ensure that the sampling methods are robust and representative to generalize the findings. The data collection process should be transparent and reproducible, with a clear explanation of how the data were collected and any measures taken to ensure consistency and accuracy.

Validity of the findings

The findings from the questionnaire analysis need to be interpreted with caution, clearly distinguishing between the validation results and the implications of the actual questionnaire responses. It is crucial to discuss how the results support (or challenge) existing theories or practices in the field of educational posture assessment.

Statistical Analysis: Provide a detailed account of the statistical methods used to analyze the data, ensuring they are appropriate for the type of data and the objectives of the study. This should include any statistical tests used, criteria for significance, and methods of handling potential data anomalies.

Additional comments

Implications and Recommendations: Discuss the practical implications of your findings for teachers, policymakers, and other stakeholders in education. Recommendations should be based on the validated data and aimed at informing future research, policy formulation, or educational practice.

Reviewer 2 ·

Basic reporting

Professional English language used throughout the paper is not clear enough, requires improvement to be understandable by the reader.
Intro & background to show context. Relevant studies in existing literature were not adequately discussed.
Line 50: The introduction requires improvement. Existing literature on the topic was not discussed at all. The authors mentioned in Line 82 “To the best of the researcher's knowledge, no scientific studies exist on educational teachers' knowledge regarding recognition, prevention, correction, and the causes of body posture defects”. However, they could have included similar studies even though the target population was different from school teachers.
Structure conforms to PeerJ standards, discipline norm, or improved for clarity.
Some figures are irrelevant.
Figure 1: How did the author compare the answers from the the test which was used as a pilot study and the retest? This procedure does not look like a test-retest reliability.
Figure 2: There is no such thing as reliability improvement of items. The aim of the test-retest reliability is to check the consistency of the responses when the same questionnaire is administered by the same participants in two different periods of time.
Figure 3: same comment as above.
Raw data supplied.

Experimental design

Original primary research within Scope of the journal.
Research question well defined, relevant & meaningful. However, rationale & benefit to literature is not clearly stated.
Investigation is not rigorous and not performed to a high technical & ethical standard.
Methods are not described with sufficient detail & information to replicate.
Lines 33-35: The authors say “..in two rounds of test-retest; The initial round encompassed 95 participants, with pre-test and post-test procedures; the second round involved 100 participants and followed a similar approach”. This is not clear. Did they do two test-retest reliability studies that is 95 participants did the pre-test and then the post-test and later 100 new participants did also a test and retest study?
The authors say in Lines 164-165 that Cohen's kappa coefficient was applied to evaluate the test-retest reliability (data from the same rater at two different points in time)”. However, this contradicts with what they mentioned in Lines 158-159 “a second round of test-retest reliability was performed with 100 newly 159 enrolled teachers who were unfamiliar with the content and structure of the questionnaire.” How is it the same rater and at the same time new participants? This is not clear.
Lines 167-169: No sufficient details, with regard to the content validity of the questionnaire, are reported. These details such as number of experts included, type of evaluation whether it is quantitative or qualitative or both and results of the validity are important to judge the validity of the instrument used.
Lines 182-193: As far as I know the test-retest reliability is a study where the same participants administer the same questionnaire at two different periods of time. However, in this study different participants did not administer the same version of the questionnaire. On the contrary, at the retest they completed a modified version of the questionnaire. How is that possible?

Validity of the findings

Impact and novelty not adequately assessed.
All underlying data have been provided however they are not robust or statistically sound.
Lines 182: The authors said “Despite our conscientious efforts to simplify the terminology used in the questionnaire, it is worth noting that the incorporation of technical and medical language relating to postural abnormalities may inadvertently have led to responses that were less reliable”. This is not true, the medical and scientific terminology is not considered to be a limitation in any research study.
The paper lacks a good discussion with comparison between the present study and previous research (Lines 238-299).
Conclusion: The authors mention in line 307 “...suitable for application in both research and practical settings”. This is too broad needs more explanation what this study adds to literature and how researchers and practitioners would benefit from this study.

Additional comments

Overall, it is hard to understand the manuscript. English language used requires improvement to make it easily understandable by the reader. It is also hard to follow the steps performed in the study. More information on how each step was done would be helpful.

---

## Round 0.2 · Minor Revisions

Thank you for providing a much improved article. I am filling in for the second reviewer as the previous could not be reached, and there are in my opinion only minor revisions required.

- L288: "as it is currently not done adequately" I think this should be toned down slightly, in the way that more is required but I am not sure anyone is to blame. See comments below (for L302-305 and even moreso L314-323 and 324-332)
- L293-301: this paragraph is choppy and difficult to read, needs some improvements.
- L302-305: the justification of the study is in the introduction. The rest of the para is also in continuation of the previous paragraph (lack of studies) and therefore should be lumped together (solution for the lack of studies in young children+geared for teachers).
- L313: the authors (plural)
- L314-323 and 324-332: these sections are overly positive and too vague, should be more neutral, and should plainly explain how this will be used (even if the entire system is not fully designed yet). The comment L288 indicates that more needs to be done, so here you should provide clear directions (for instance, what resources should teachers and administrators reading this should turn to?).

Reviewer 1 ·

Basic reporting

The changes have significantly improved the article and now it is understandable and clear

Experimental design

it has improved

Validity of the findings

The same of the other sections

Additional comments

The paper can be accepted in this form

---

## Round 0.3 · accepted · Accept

Thank you for providing the last round of amendments. Since the expert reviewer already gave their go-ahead at the previous step, I'll accept without sending to reviewers. There are still very minor corrections but feel this should be taken care of at the proof stage, as these have no bearing on the academic value of the article.

in the new doc (not the track-changed one):
- sentence L274-275 has twice questionnaire (one should probably just be "items") and twice reliability. Check for legibility.
- L 292: "However, there are no questionnaires"...
- L 299: please check, for me this would be "postural defects and deformities"?